# Photodestruction of Diatomic Molecular Ions: Laboratory and Astrophysical Application

**DOI:** 10.3390/molecules26010151

**Published:** 2020-12-31

**Authors:** Vladimir A. Srećković, Ljubinko M. Ignjatović, Milan S. Dimitrijević

**Affiliations:** 1Institute of Physics Belgrade, Belgrade University, Pregrevica 118, 11080 Belgrade, Serbia; ljuba@ipb.ac.rs; 2Astronomical Observatory, Volgina 7, 11060 Belgrade, Serbia; mdimitrijevic@aob.rs; 3Observatoire de Paris, PSL Université, CNRS, Sorbonne Université, F-92195 Meudon, France

**Keywords:** photodissociation, diatomic molecules, plasma diagnostics, astrophysical processes, radiative processes, molecular spectroscopy, planetary chemistry, planetary geochemistry, atomic and molecular data

## Abstract

In this work, the processes of photodissociation of some diatomic molecular ions are investigated. The partial photodissociation cross-sections for the individual rovibrational states of the diatomic molecular ions, which involves alkali metals, as well as corresponding data on molecular species and molecular state characterizations, are calculated. Also, the average cross-section and the corresponding spectral absorption rate coefficients for those small molecules are presented in tabulated form as a function of wavelengths and temperatures. The presented results can be of interest for laboratory plasmas as well as for the research of chemistry of different stellar objects with various astrophysical plasmas.

## 1. Introduction

Advances in the research of ultra-low-temperature gases have prompted a new interest in alkali–dimer cations (see e.g., [1]). Recently, potential energy curves of some diatomic molecules have been investigated with a special focus both on neutral and ionized alkali–dimer systems [2,3,4]. Calculations of the potential energy curves for molecules considered here have already been published: LiH${}^{+}$ and NaH${}^{+}$ in [5], Li${}_{2}^{+}$ and Na${}_{2}^{+}$ in [6] and for LiNa${}^{+}$ in [7]. These data are given in the Appendix A. This particular concern is generally due to the recent development in cold collision dynamics involving alkali diatomic systems.

From [8,9,10] and from investigation in existing literature about the solar photospheric spectra, the absorption spectra of planetary atmospheres and the chemical composition of dust, it follows that alkali metals take part in the composition of cosmic objects [11,12]. It has been discovered and concluded that important characteristics of the brown dwarfs and extrasolar planets spectra might be attributed to resonance lines of alkali metal atoms broadened by collisions with ambient helium atoms and hydrogen molecules (see e.g., [13,14]). Alkali metal resonance lines shape the optical spectrum from the UV to the near-IR spectral range [13,15].

Most alkali hydride ionic and neutral molecules are very useful in understanding the formation and the evolution of the Molecular Universe [16]. Even though they are involved in various astrophysical and astrochemical processes as radiative transfer or dissociative recombination for instance [17], their spectroscopy is practically unknown both theoretically and experimentally, in particular in the case of molecular ions [18]. The charge transfer in collision of alkali atoms with protons affects the ionization balance in the atmospheres of various environments such as planets, dwarf stars, and the interstellar medium, and is of interest in the area of plasma fusion as a method of energetic neutral beam injection in fusion reactors [19]. Also, one can note the current importance of investigation of optical properties of various small molecules (see e.g., [20,21,22,23,24]) and corresponding atomic and molecular (A & M) data.

In this paper, we will focus on the investigation of photodestruction of the following species: ${\mathrm{LiH}}^{+}$, ${\mathrm{NaH}}^{+}$, ${\mathrm{Li}}_{2}^{+}$, ${\mathrm{Na}}_{2}^{+}$ and ${\mathrm{LiNa}}^{+}$. Here we will examine the significance of the processes of molecular-ion photodissociation in symmetric case:(1)$$h\nu +M_{2}^{+}⟹M+M^{+},$$
where *M* and $M^{+}$ are alkali atom and ion in their ground states, and $M_{2}^{+}$ is molecular-ion in the ground electronic state. Such processes are photodissociation of Li${}_{2}^{+}$, Na${}_{2}^{+}$ but also LiNa${}^{+}$ ions, keeping in mind a small difference between the ionization energies of Li and Na atoms. Also, we will examine the non-symmetric case for the processes of photodissociation:(2)$$h\nu +MX^{+}⟹X^{+}+M,$$
where *M* is an atom whose ionization potential is less than the corresponding value for atom *X*. $MX^{+}$ is also molecular-ion in the ground electronic state. The photodissociation of HLi${}^{+}$ and HNa${}^{+}$ ions is just an example of strongly non-symmetric processes, keeping in mind that the ionization energy of hydrogen is much higher than the ionization energy of Li and Na atoms. Processes (1,2) are schematically shown in Figure 1 by solid arrows. By absorbing photons of the same energy molecular-ion, MX${}^{+}$ can pass into one of the higher excited electronic states (dashed arrow) but the probability of these transitions is small compared to the probability of resonant processes (1,2). That is why the influence of the processes represented by the dashed arrow in Figure 1 will not be considered in this paper.

We determine theoretically calculated partial photodissociation cross-sections for the individual rovibrational states of the diatomic molecular ions ${\mathrm{LiH}}^{+}$, ${\mathrm{NaH}}^{+}$, ${\mathrm{Li}}_{2}^{+}$, ${\mathrm{Na}}_{2}^{+}$ and ${\mathrm{LiNa}}^{+}$ as well as the corresponding data on molecular species and molecular state characterizations (rovibrational energy states, etc.). The data are obtained for ${\mathrm{Li}}_{2}^{+}$, ${\mathrm{Na}}_{2}^{+}$ and ${\mathrm{LiNa}}^{+}$, for 500 K≤T≤5000 K and 400 nm≤λ≤1000 nm (for modeling laboratory plasmas and geo-cosmic weakly ionized plasma) and for ${\mathrm{LiH}}^{+}$, ${\mathrm{NaH}}^{+}$ for extended temperature range 100 K≤T≤ 20,000 K and wavelength range 60 nm≤λ≤160 nm (for additional astro usage in modeling Solar photosphere, atmosphere above sunspots and Lithium-Rich Stars).

Also, the average cross-section and the corresponding spectral absorption rate coefficients for the photodissociation of the mentioned molecular ions are obtained for a wide region of wavelengths and temperatures. The presented results here are ready for further use with a particular emphasis on the applications for astro-plasma research and low-temperature laboratory plasma research [10,13,25,26,27].

The paper is structured as follows. In Section 3 we describe the methods of calculation and calculated quantities, in Section 2 we present the results of the calculation and discussed results and in Section 4 the conclusions are presented with further directions of research.

## 2. Results and Discussion

The partial photodissociation cross-sections for the individual rovibrational states of the diatomic molecular ions which involve alkali metals are calculated. Also, the average cross-section and the corresponding spectral absorption rate coefficients for those small molecules ${\mathrm{LiH}}^{+}$, ${\mathrm{NaH}}^{+}$, ${\mathrm{Li}}_{2}^{+}$, ${\mathrm{Na}}_{2}^{+}$ and ${\mathrm{LiNa}}^{+}$ are obtained for a wide region of wavelengths and temperatures. The data are obtained for ${\mathrm{Li}}_{2}^{+}$, ${\mathrm{Na}}_{2}^{+}$ and ${\mathrm{LiNa}}^{+}$, for 500 K≤T≤5000 K and 400 nm≤λ≤1000 nm (for modeling laboratory plasmas and geo-cosmic weakly ionized plasma) and for ${\mathrm{LiH}}^{+}$, ${\mathrm{NaH}}^{+}$ for extended temperature range 100 K≤T≤ 20,000 K and wavelength range 60 nm≤λ≤160 nm (for additional astro usage in modeling Solar photosphere, atmosphere above sunspots and Lithium-Rich Stars). The results are presented in this section by figures and table and in Appendix A.

### 2.1. Cross-Sections

In Figure 2a,b and Figure 3a,b we present a surface plot of the averaged cross-section σ(λ,T) for photodissociation of the ${\mathrm{LiH}}^{+}$, ${\mathrm{NaH}}^{+}$, ${\mathrm{Li}}_{2}^{+}$ and ${\mathrm{LiNa}}^{+}$ molecular ions as a function of λ for a wide range of temperatures *T*, which are relevant for modeling laboratory [13] and astrophysical plasmas (e.g., geo-cosmic plasmas) [10,11,28]. These figures and the shape of surfaces show that exist a noticeable difference between temperature dependence of the mean thermal photoionization cross-section for different species. Figure 2a,b show that in the cases of ${\mathrm{LiH}}^{+}$, ${\mathrm{NaH}}^{+}$, σ(λ,T) in the region 110 nm ≤λ≤ 160 nm significantly increases and becomes noticeable, while decreases for λ outside this region. One can notice that the cross-section sudden increase at lower temperature for these cases. Opposite behavior are presented in Figure 3a,b. These figures show that in the cases of ${\mathrm{Li}}_{2}^{+}$ and ${\mathrm{LiNa}}^{+}$, σ(λ,T) significantly increases in the region 400 nm ≤λ≤ 700 nm and becomes noticeably and decreasing for λ≥700 nm. Also, the temperature dependance is complex.

The obvious qualitative difference in the photodissociation cross-section behavior shown in Figure 2 and Figure 3 can be explained by the fact that they refer to two cases sharply separated at the beginning: Figure 2 refers to highly non-symmetric processes of photodissociation of HLi${}^{+}$ and HNa${}^{+}$, while Figure 3 refers to the symmetric case of photodissociation of Li${}_{2}^{+}$, Na${}_{2}^{+}$ and LiNa${}^{+}$.

### 2.2. Rate Coefficients

The behavior of molecular-ion photodissociation rate coefficient K(λ,T), calculated by Equation (Equation 6), is graphically presented in Figure 4. In order to enable wider use of these results in various stellar models [28,29], examples of the NaH${}^{+}$ and LiH${}^{+}$ for temperature range 500 K ≤T≤ 20,000 K and for different wavelengths are studied. The data are presented in Appendix A by tables for all species. This allows direct calculation of the spectral absorption coefficients during the process of applying an atmosphere model with the given parameters and composition of plasma.

To prepare easier and more satisfying usage of calculated data in modeling as well as in explanation of experimental results in laboratories, we give a simple fitting formula for the photodissociation rate coefficients as a second-degree polynomial (logarithmic) based on a least-square method:(3)$$\log\left(K\left(\lambda ;T\right)\right)=\sum_{i=0}^{2}c_{i}\left(\lambda \right){\left(\log\left(T\right)\right)}^{i}.$$

In Table 1 are listed the coefficients $c_{i}\left(\lambda \right)$ i.e., selected fits for the photodissociation rate coefficients. The fitting formula is valid within the temperature range 500 K≤T≤5000 K and wavelengths 400 nm≤λ≤1000 nm.

Also, we give table in online material with the data of the selected fits of the Equation (Equation 3) to the photodissociation rate coefficient for LiH${}^{+}$, NaH${}^{+}$.

### 2.3. Modeling

The calculated rate coefficient (see Table 1 and Appendix A) for the presented plasma parameters could be of interest for the models of Io’s atmosphere [30]. This was poorly investigated in this sense till now and consequently, such processes may become important and could be used for better numerical simulations and modeling. In Figure 5 (lower panel) the temperature and density altitude profiles for low density, moderate density and high density of Io modeled atmospheres [30] are presented. The upper panels of Figure 5 shows the photodissociation rate coefficients K(λ,T) on the example of Na${}_{2}^{+}$, Li${}_{2}^{+}$ and LiNa${}^{+}$ for modeled atmospheres of Io [30,31]. One can notice the growth of coefficients at lower altitudes. Additionally, one can see that the rate coefficients are the largest around 600 nm for Na${}_{2}^{+}$ while for LiNa${}^{+}$ at 500 nm and at 400 nm for Li${}_{2}^{+}$.

The additional examples i.e., the figures for the moderate-density and high-density models of Io’s atmosphere can be found in Appendix A. We can note that the investigation of these processes is needed for better understanding of the interaction between the Io’s atmosphere and Jovian magnetosphere [32] and of the processes in the Jovian environment [33]. Even though lithium has not been spectroscopically discovered at Io till now, some scientists expect its presence [34], and for the estimation of the abundances of lithium as well as of other candidate elements such as fluorine, rubidium, caesium, etc. at Io, the corresponding spectral coefficients are needed. Also, the data for lithium could be of interest in modeling of cold stars with low effective temperatures and enormous high Li abundances (e.g., subgiant J0741+2132, see [35]).

### 2.4. Further Usage

We prepared data in Virtual Observatory (VO) (https://www.ivoa.net/documents) compatible format [36,37,38] which enable further implementation of results in the A & M databases like [27,39] thus providing a single location where users can access atomic and molecular data. Also, in order to prepare the easier use of data in practice, we found and give for the rate coefficients a simple and accurate fitting formula as a second-degree polynomial fit to numerical results.

### 2.5. Development

The next step of development is to modify and adapt presented method to enable theoretical research of photodestruction processes which involves larger and complicated molecules for further exploration and explanation of experimentally obtained results e.g., in synchrotron facility (see research of [40,41,42]) for potential industry, biological, astrobiological and astrochemistry applications [43,44].

## 3. The Method and Calculated Quantities

The rate coefficients and cross-sections are acquired using a quantum mechanical method where the photodissociation process is treated as outcome of radiative transitions between the ground and the first excited adiabatic electronic state of the molecular-ion (see e.g., [45]). The transitions are the result of the interaction of the ion-atom systems electronic component with the electromagnetic field in the dipole approximation.

### 3.1. Photodissociation Cross-Sections

In accordance with the dipole approximation the partial cross-sections $\sigma_{J,v}\left(\lambda \right)$ for the rovibrational states with specified quantum numbers *J* and *v*, are given in accordance with [46,47] by expression
(4)$$\begin{array}{c} \sigma_{J,v}\left(\lambda \right)=\frac{8\pi^{3}}{3\lambda }[\frac{J+1}{2J+1}{\left|D_{J,v;J+1,E_{imp}^{\prime }}\right|}^{2}\\ +\frac{J}{2J+1}|D_{J,v;J-1,E_{imp}^{\prime }}{|}^{2}], \end{array}$$
where $D_{J,v;J\pm 1,E_{imp}^{\prime }}=$
$<in,J,v;R|D_{in,fin}\left(R\right)|fin,J\pm 1,E^{\prime }>$. $D_{in;fin}\left(R\right)$ is the electronic dipole matrix element $D_{in;fin}\left(R\right)=<in;R\left|D\left(R\right)\right|fin;R>,$ where D is the operator of the dipole moment of the considered system. Here $E^{\prime }=\epsilon_{J,v}+\varepsilon_{\lambda }$, $E_{imp}^{\prime }$ and $E^{\prime }$, $D_{in,fin}\left(R\right)$ and the states are given in [45]. Figure 6 present illustrative example for $\sigma_{J,v}\left(\lambda \right)$ defined by Equation (Equation 4) for the case J=0 and v=10, in the wavelength region 50nm≤λ≤1500nm for hydrogen diatomic molecular-ion (see [47]).

In the general case mean thermal photodissociation cross-section can be expressed as in [46,48] by the relations
(5)$$\sigma \left(\lambda ,T\right)=\frac{\sum_{J,v}\left(2J+1\right)e^{\frac{-E_{J,v}}{kT}}\cdot \sigma_{J,v}\left(\lambda \right)}{\sum_{J,v}\left(2J+1\right)e^{\frac{-E_{J,v}}{kT}}},$$
where $E_{J,v}$ designate the energies of states with the respect to the ground rovibrational states. In Equation (Equation 5) $E_{J,v}=E_{dest}+\varepsilon_{J,v}$, where $E_{dest}$ is the dissociative energy of the molecular-ion, and the energies $\varepsilon_{J,v}<0$ are defined as in [45,47].

### 3.2. Photodissociation Spectral Rate Coefficient

Using Equation (Equation 5) the photodissociation spectral rate coefficient can be determined as in [46] by the expression
(6)$$K\left(\lambda ,T\right)=\sigma \left(\lambda ,T\right)\cdot \zeta^{-1}\left(T\right),$$
where the factor $\zeta \left(T\right)=N\left(X\right)N\left(M^{+}\right)/N\left(MX^{+}\right)$ in Equation (Equation 6) is given by the relation
(7)$$\zeta \left(T\right)=\frac{g_{1}g_{2}}{g_{12}}{\left(\frac{\mu kT}{2\pi \hbar^{2}}\right)}^{\frac{3}{2}}\cdot \frac{1}{\sum_{J,v}\left(2J+1\right)e^{\frac{E_{dest}-E_{J,v}}{kT}}}.$$

Here $g_{12}$, $g_{1}$ and $g_{2}$ are the electronic statistical weights of the species, $MX^{+}$, *X* and $M^{+}$ respectively, and σ(λ,T) is given by Equation (Equation 5). N(X), $N(M^{+})$ and $N(MX^{+})$ are density of the *X*, $M^{+}$ and $MX^{+}$, respectively.

This enables further calculation of the spectral absorption coefficients during the process of applying an atmosphere model with the given plasma parameters or e.g., plasma composition in laboratory investigation. The bound-free absorption process (Equation 1) i.e., processes of diatomic molecular-ion photodissociation can be characterized by partial spectral absorption coefficients κ(λ) (see e.g., [26]). It can be presented by relation $\kappa \left(\lambda \right)=\sigma \left(\lambda ,T\right)N\left(M_{2}^{+}\right)=K\left(\lambda ,T\right)N\left(M\right)N\left(M^{+}\right)$ where $N(M_{2}^{+})$ is density of the $M_{2}^{+}$ and σ(λ,T) is average cross-section for photodissociation of this molecular-ion Equation (Equation 5) and the coefficient K(λ,T) defined by Equation (Equation 6). Correspondingly, with the definition of the absorption coefficient, the coefficient K(λ,T) is given in units cm${}^{5}$. For the non-symmetric case (Equation 2) i.e., $MX^{+}$ the relations are similar.

## 4. Conclusions

We have calculated the average cross-section for the photodissociation processes and the corresponding spectral rate coefficient of the diatomic molecular ions ${\mathrm{LiH}}^{+}$, ${\mathrm{NaH}}^{+}$, ${\mathrm{Li}}_{2}^{+}$, ${\mathrm{Na}}_{2}^{+}$ and ${\mathrm{LiNa}}^{+}$ for overall range of 500K≤T≤20000 K and 60nm≤λ≤1000nm. We present the rate coefficients of the corresponding processes in tabulated form, which is appropriate for further use. The obtained results have potential astrophysical use in the improvement of chemistry and modeling of different layers of weakly ionized plasmas e.g., in the atmospheres of various stars and have also been applied to models of the atmosphere of the Io. The results are also important in theoretical and laboratory spectroscopic plasmas research, industry, and technology application.

## Figures and Tables

**Figure 1 molecules-26-00151-f001:**
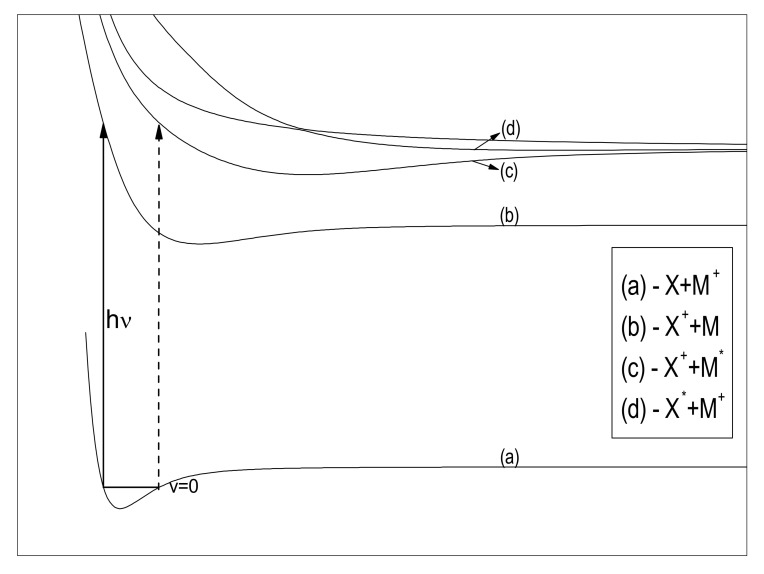
*X* and *M* denote atoms in the ground state, $X^{*}$ and $M^{*}$—atoms in the first excited state and $X^{+}$ and $M^{+}$—atomic ions. Solid arrow: The molecular-ion $MX^{+}$ in the ground state (**a**) absorbs the photon and passes into the first excited electronic state (**b**) which decays into the ion $X^{+}$ and the *M* atom in the ground state. Dashed arrow: Molecular-ion $MX^{+}$ absorbs a photon and passes into one of the higher electronic states (**c**,**d**) and decays so that one of the atoms remains in the excited state.

**Figure 2 molecules-26-00151-f002:**
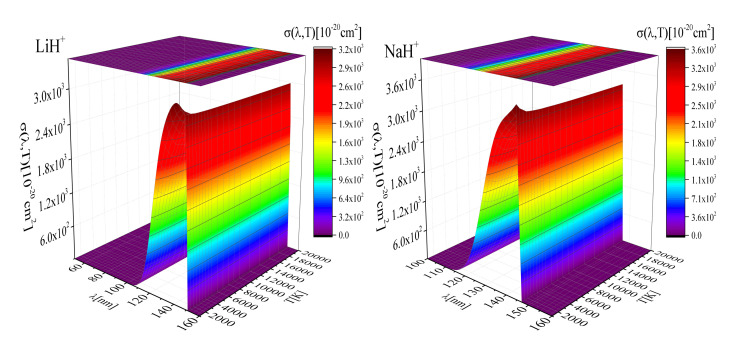
**Left**: A surface plot of the averaged cross-section σ(λ,T) for photodissociation of the ${\mathrm{LiH}}^{+}$ molecular-ion, as a function of λ and *T*; **Right**: A surface plot of the averaged cross-section σ(λ,T) for photodissociation of the ${\mathrm{NaH}}^{+}$ molecular-ion, as a function of λ and *T*.

**Figure 3 molecules-26-00151-f003:**
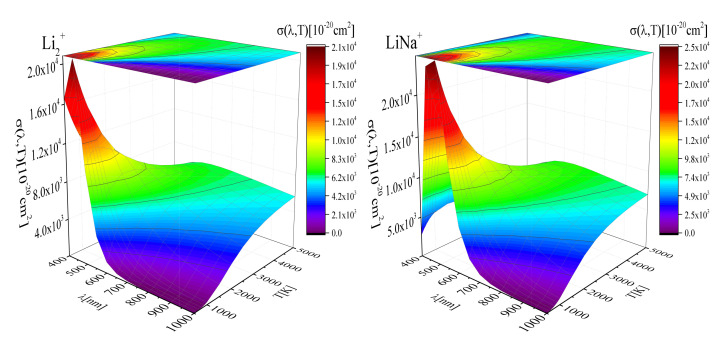
**Left**: A surface plot of the averaged cross-section σ(λ,T) for photodissociation of the ${\mathrm{Li}}_{2}^{+}$ molecular-ion, as a function of λ and *T*; **Right**: A surface plot of the averaged cross-section σ(λ,T) for photodissociation of the ${\mathrm{LiNa}}^{+}$ molecular-ion, as a function of λ and *T*.

**Figure 4 molecules-26-00151-f004:**
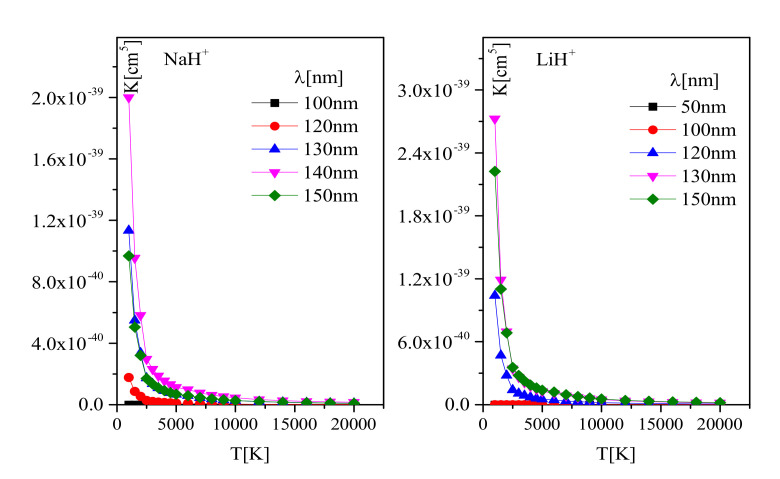
The plot of the spectral rate coefficients K(λ,T) for photodissociation of the NaH${}^{+}$, LiH${}^{+}$, as a function of *T* and λ.

**Figure 5 molecules-26-00151-f005:**
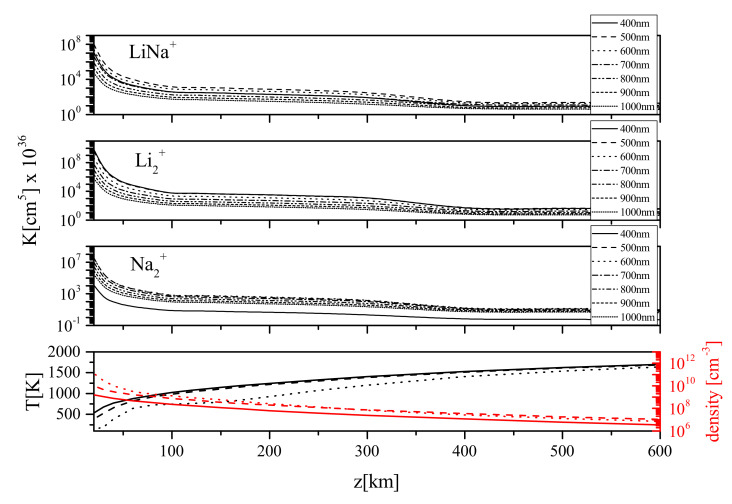
Lower panel: temperature and density altitude profiles for low-density (solid line), moderate-density (dashed) and high-density (dotted line) models of Io’s atmosphere. Upper panels: the total photodissociation rate coefficient for Na${}_{2}^{+}$, Li${}_{2}^{+}$ and LiNa${}^{+}$ for an example of the low-density model of Io’s atmosphere [30].

**Figure 6 molecules-26-00151-f006:**
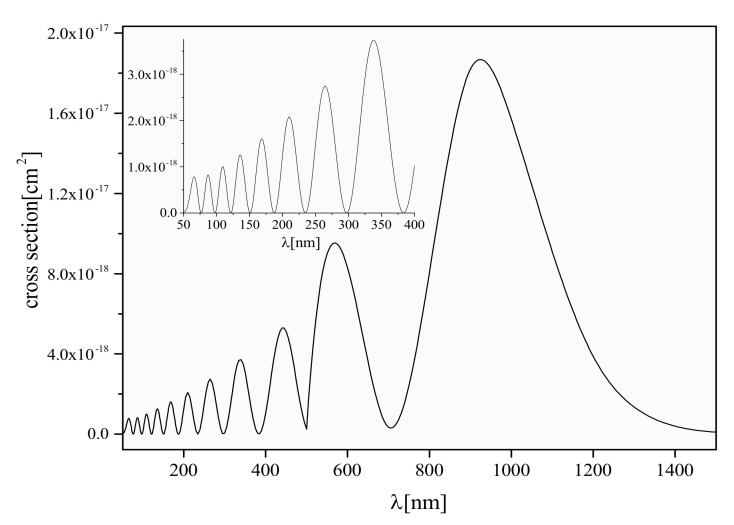
The behavior of the cross-section $\sigma_{J,v}\left(\lambda \right)$ Equation (Equation 4) for J=0 and v=10, as a function of λ (see [47]).

**Table 1 molecules-26-00151-t001:** Data of the selected fits of the Equation (Equation 3) to the photodissociation rate coefficient for ${\mathrm{Li}}_{2}^{+}$, ${\mathrm{Na}}_{2}^{+}$ and ${\mathrm{LiNa}}^{+}$.

	${\mathrm{Li}}_{2}^{+}$			${\mathrm{Na}}_{2}^{+}$			${\mathrm{LiNa}}^{+}$		
λ [nm]	$c_{0}$	$c_{1}$	$c_{2}$	$c_{0}$	$c_{1}$	$c_{2}$	$c_{0}$	$c_{1}$	$c_{2}$
400	115.794	−83.441	11.3946	41.7324	−43.2074	5.88338	75.7089	−61.4123	8.36583
500	110.628	−80.5359	10.9971	82.4755	−65.1067	8.8713	85.8781	−66.8371	9.10941
600	97.0323	−73.0543	9.96852	80.6419	−64.0402	8.72452	76.0451	−61.3987	8.36012
700	84.114	−65.944	8.98853	72.9253	−59.7686	8.13487	64.6813	−55.126	7.49352
800	72.9344	−59.7847	8.13795	64.6227	−55.1811	7.50068	54.2741	−49.373	6.69625
900	63.4083	−54.5306	7.41114	56.8894	−50.9076	6.90904	45.2003	−44.3536	5.99965
1000	55.2672	−50.0354	6.7884	49.9549	−47.0722	6.37719	37.3453	−40.0034	5.39484

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
