# Peer review of "Photodestruction of Diatomic Molecular Ions: Laboratory and Astrophysical Application"

_molecules, 2020, doi:10.3390/molecules26010151_

Round 1
Reviewer 1 Report
Reviewer's comments
Manuscript ID: molecules-1057119
Type of manuscript: Regular Article
Title: Photodestruction of diatomic molecular ions:
laboratory and astrophysical application
Authors: Vladimir A. Sreckovic, Ljubinko M. Ignjatovic, Milan S. Dimitrijevic
In this paper, the authors investigated the processes of photodissociation of some diatomic molecular ions. The partial photodissociation cross-sections and the corresponding spectral absorption rate coefficients for those small molecules are obtained for the wide region of plasma parameters. Authors presented the rate coefficients of the corresponding processes in tabulated form, which is very usefull for further use. The presented results can be of interest in theoretical and laboratory spectroscopic plasmas research, industry and technology application as well as for modeling various astrophysical plasma.
The results are original. The manuscript is well organized and clearly written. I believe that this research will be interesting and useful for the readers in the field. I would like to recommend publication of this paper in journal Molecules.
Some minor remarks:
Page3,line72: make it uniform like in the text i.e. photo-dissociation=> photodissociation. Same goes for photo-ionization => photoionization (p4,l87).
Page 6, l 105: “The calculated rate coefficient (see Tab. 1)…” => “The calculated rate coefficient (see Tab. 1 and Supplementary material)…”
It is not mandatory, but I think that manuscript can be improved by including additional details about importance of results about molecules with lithium in astrophysical object like Lithium-Rich Stars e.g. J0741+2132. It would be good if authors could insert a few sentences at the end of the Section 3.3.
Author Response
Answer: Thank you. We appreciate your positive opinion about our manuscript and also for your valuable comments and suggestions. We inserted suggested corrections.

Reviewer 2 Report
This is a well written manuscript which presents interesting results that will be useful for potential users. Small changes and corrections are suggested to make the content easier for the reader to understand. Comments are included in the pdf file attached.

Author Response
Answer: We appreciate your positive opinion about our manuscript. We thank the Reviewer for careful reading and useful comments. We inserted corrections as suggested.

Reviewer 3 Report
In this manuscript, the authors calculated the average cross-section, the spectral rate coefficient for the photodissociation processes of some diatomic molecular ions across a broad wavelength and temperature range. However, the authors just presented the calculation results with little to no explanation. A major revision is needed. See below. 1. For equation 1 and 2, the authors setup the equation and assume that all atoms or ions are in their electronic ground state. However, I am assuming that after the molecule ion absorbs the photon, it must be excited to the excited state, then dissociates, then relax to the ground state. Is this the case? If so, this is a rather complicated process, can the author’s calculation accurately simulate the process? 2. Figure 2, it looks like temperature has very little effect on the cross-section comparing with the wavelengths, what is the reason? Also, it looks like the absorption peak for NaH+ is at 140 nm while it is at 130 nm for LiH+, what causes the difference? 3. Figure 3, why in this case, both wavelengths and temperatures have such large effects on the cross-section? 4. Table 1. Why not fit the rate coefficients for LiH+ and NaH+. 5. Line 112-113, the authors said "rate coefficients are the largest around 600 nm" while it shows 500 nm for LiNa+, 400 nm for Li2+ and it is very hard to see for Na2+ based on the upper panel of Figure 5. Also, why use the low-density model? What are the differences between the three models?Author Response
We thank the referee for the report. Corrections have been done according to the comments, changes have been marked. We modified/improved the text, add additional figure (Fig 1.) in the text, add several references, and in Supplementary material we added 6 figures and table.
In this manuscript, the authors calculated the average cross-section,
the spectral rate coefficient for the photodissociation processes of
some diatomic molecular ions across a broad wavelength and temperature
range. However, the authors just presented the calculation results
with little to no explanation. A major revision is needed. See below.
1. For equation 1 and 2, the authors setup the equation and assume
that all atoms or ions are in their electronic ground state. However,
I am assuming that after the molecule ion absorbs the photon, it must
be excited to the excited state, then dissociates, then relax to the
ground state. Is this the case? If so, this is a rather complicated
process, can the author’s calculation accurately simulate the process?
Answer: Concerning this point, in order to better explain investigated processes, we added the sentence at the end of the first paragraph of Introduction “Calculations of the potential energy curves for molecules considered here have already been published: 16 … in [7].”
Also, we added new Fig 1. and the part of the text just after Eq.2 “The photodissociation of …be considered in this paper.”.
Figure 2, it looks like temperature has very little effect on the
cross-section comparing with the wavelengths, what is the reason?
Also, it looks like the absorption peak for NaH+ is at 140 nm while it
is at 130 nm for LiH+, what causes the difference? 3. Figure 3, why in
this case, both wavelengths and temperatures have such large effects
on the cross-section?
Answer: Concerning this point we added the sentence after Eq.1 “Such processes are photodissociation of …, keeping in mind a small difference between the ionization energies of Li and Na atoms.” and after Eq 2. “The photodissociation of HLi+ and HNa+ ions is just an example of strongly non-symmetric processes .. atoms.”
Also, we added whole paragraph at the end of the Subsec.3.1. “The obvious qualitative difference in the photodissociation cross-section behavior shown in Figs. 3 and 4… LiNa+.”
- Table 1. Why not fit the rate coefficients for LiH+ and NaH+. 5.
Answer: Thank you, for this suggestion. The new table with fits of the rate coefficients for LiH+ and NaH+ is provided in the Supplement material in order not to load the text. We added the part of the text concerning this at the end of Subsect. 3.2: “… Also we give table in online material with the data of the selected fits … rate coefficient for LiH$^{+}$, NaH$^{+}$. …”
Line 112-113, the authors said "rate coefficients are the largest around 600 nm" while it shows 500 nm for
LiNa+, 400 nm for Li2+ and it is very hard to see for Na2+ based on
the upper panel of Figure 5.
Answer: Corrected. We enlarged Fig 5 (now Fig 6) and corrected text as suggested. See sentence in subsec 3.3 “Additionally, one can see that the rate coefficients are the largest around 600 nm for Na$_{2}^{+}$, at 500 nm for LiNa$^{+}$ and at 400 nm for Li$_{2}^{+}$.”
Also, why use the low-density model? What are the differences between the three models?
Answer: In principle, the differences in the models are in the plasma parameters, but nothing drastic. We wanted to emphasize in the text the importance of the process in general for Io modeling with the one example (we chose low-density). Of course, if someone is interested, we additionally give the Figs. with moderate-density and high-density and they are in the Supplementary material.
Also, we add the part of the text concerning this: “… The additional examples i.e. the figures for the moderate-density and high-density … material.…”

Round 2
Reviewer 3 Report
The authors have addressed my concerns and the paper can be published in its present form.